# Design, Synthesis, and Antiproliferative Activity of New 5-Chloro-indole-2-carboxylate and Pyrrolo[3,4-*b*]indol-3-one Derivatives as Potent Inhibitors of EGFR^T790M^/BRAF^V600E^ Pathways

**DOI:** 10.3390/molecules28031269

**Published:** 2023-01-28

**Authors:** Lamya H. Al-Wahaibi, Anber F. Mohammed, Mostafa H. Abdelrahman, Laurent Trembleau, Bahaa G. M. Youssif

**Affiliations:** 1Department of Chemistry, College of Sciences, Princess Nourah bint Abdulrahman University, Riyadh 11564, Saudi Arabia; 2Pharmaceutical Organic Chemistry Department, Faculty of Pharmacy, Assiut University, Assiut 71526, Egypt; 3Pharmaceutical Organic Chemistry Department, Faculty of Pharmacy, Al-Azhar University, Assiut Branch, Assiut 71524, Egypt; 4School of Natural and Computing Sciences, University of Aberdeen, Meston Building, Aberdeen AB24 3UE, UK

**Keywords:** indole, pyrrole, mutant EGFR, BRAF^V600E^, melanoma, anticancer

## Abstract

Mutant EGFR/BRAF pathways are thought to be crucial targets for the development of anticancer drugs since they are over-activated in several malignancies. We present here the development of a novel series of 5-chloro-indole-2-carboxylate **3a–e**, **4a–c** and pyrrolo[3,4-*b*]indol-3-ones **5a–c** derivatives as potent inhibitors of mutant EGFR/BRAF pathways with antiproliferative activity. The cell viability assay results of **3a–e**, **4a–c**, and **5a–c** revealed that none of the compounds tested were cytotoxic, and that the majority of those tested at 50 µM had cell viability levels greater than 87%. Compounds **3a–e**, **4a–c**, and **5a–c** had significant antiproliferative activity with GI_50_ values ranging from 29 nM to 78 nM, with **3a–e** outperforming **4a–c** and **5a–c** in their inhibitory actions against the tested cancer cell lines. Compounds **3a–e** were tested for EGFR inhibition, with IC_50_ values ranging from 68 nM to 89 nM. The most potent derivative was found to be the *m*-piperidinyl derivative **3e** (R = *m*-piperidin-1-yl), with an IC_50_ value of 68 nM, which was 1.2-fold more potent than erlotinib (IC_50_ = 80 nM). Interestingly, all the tested compounds **3a–e** had higher anti-BRAF^V600E^ activity than the reference erlotinib but were less potent than vemurafenib, with compound **3e** having the most potent activity. Moreover, compounds **3b** and **3e** showed an 8-fold selectivity index toward EGFR^T790M^ protein over wild-type. Additionally, molecular docking of **3a** and **3b** against BRAF^V600E^ and EGFR^T790M^ enzymes revealed high binding affinity and active site interactions compared to the co-crystalized ligands. The pharmacokinetics properties (ADME) of **3a–e** revealed safety and good pharmacokinetic profile.

## 1. Introduction

Cancer has been a major public health issue around the world, with an increasing number of patients diagnosed each year [1]. Unfortunately, chemotherapy’s effectiveness as a primary mode of cancer treatment is hampered by drug resistance, severe side effects, and poor selectivity [2,3]. Thus, recently, immunotherapy and newly combined, multi-targeted therapies have been recommended [4,5,6]. Kinase activation in various cell signaling pathways has been linked to cancer cell survival, invasiveness, and drug resistance [7,8]. As a result, anticancer drugs that target kinases, such as the epidermal growth factor receptor (EGFR) and serine/threonine kinases, such as BRAF, are gaining popularity [9,10].

RAF mutations are found in roughly 70% of melanoma, 100% of hairy cell leukemia, and 41% of hepatocellular carcinoma. Meanwhile, EGFR mutations such as T790M and C797S have been identified as important therapeutic targets in lung, breast, and epithelial cancers [11,12,13]. Mutant RAF/EGFR pathways are over-activated in a variety of cancers, and they are regarded as critical targets for anti-cancer drug development [14,15,16]

On the other hand, one of the most well-known structures with robust anticancer activity is the indole skeleton, which is present in many active substances and natural products [17]. To date, numerous indole derivatives have been discovered to be effective anticancer agents; some of them have even been used in clinics [18,19,20]. In the literature research, several indole-based compounds with tyrosine kinase inhibitory action have been reported [21,22,23]. Compound **I** (Figure 1) was reported to have potent anticancer activity against four cancer cell lines, as well as promising EGFR inhibitory activity [21]. Compound **II** has been identified as a dual EGFR^T790M^/c-MET inhibitor capable of targeting resistant NSCLC [22]. Compound **II** had IC_50_ values of 0.094, 0.099, and 0.595 µM for EGFR^T790M^, EGFR^L858R^, and c-MET, respectively (Figure 1). Osimertinib, an indole-based drug, (**III**, Figure 1) is an EGFR TKI with a 200-fold selectivity index toward EGFR T790M/L858R protein over wild-type EGFR [23]. Osimertinib was approved by the FDA in 2015 to treat EGFR T790M-positive NSCLC [23]. Recently [24], we reported on the development of a novel series of 5-chloro-3-hydroxymethyl-indole-2-carboxamides **IVa-l** (Figure 1) as EGFR-TK antiproliferative agents. Compound **IVc** (R= 4-morpholin-4-yl) was the most potent EGFR inhibitor, with an IC_50_ value of 0.12 µM.

A series of pyrazino[1,2-*a*]indol-1(2*H*)-ones has been reported [25] as antiproliferative agents targeting EGFR and BRAF^V600E^. Compound **V** (Figure 1) inhibits both EGFR and BRAF^V600E^ with IC_50_ values of 1.7 µM and 0.1 µM, respectively [25]. Following this, a series of structural modifications to our lead compound **V** to design and synthesize a new series of pyrazino[1,2-*a*]indol-1(2*H*)-ones [26]. Compound **VI (**Figure 1**)** was the most effective derivative, with a GI_50_ value of 1.107 µM against four cancer cell lines. **VI** inhibited EGFR with an IC_50_ of 0.08 µM but only moderately inhibited BRAF^V600E^ with an IC_50_ of 0.15 µM. In another study [27], we describe our efforts to synthesize and optimize a novel class of potent antiproliferative agents **VII** (Figure 1). The antiproliferative activity of the target compounds is impressive. These compounds have a dual inhibitory effect on EGFR and BRAF^V600E^, with IC_50_ values of 32 nM and 45 nM, respectively.

Motivated by the data presented above, and as part of our ongoing efforts to identify promising lead compounds for dual or multi-targeted anticancer agents [28,29,30], we present herein the design and synthesis of a novel class of indole-2-carboxylates, compounds **3a–e** and **4a–c** (Scaffold A), as well as 1,2-dihydropyrrolo[3,4-*b*]indol-3(4*H*)-ones, compounds **5a–c** (Scaffold B) (Figure 1), as dual EGFR/BRAF^V600E^ inhibitors with antiproliferative activity. The new compounds will be evaluated for their safety profile by assessing their effect on the viability of human normal cell lines, while their antiproliferative activity will be evaluated against a panel of four cancer cell lines. The most potent compounds will be evaluated for their ability to inhibit wild-type EGFR (EGFR^WT^) and BRAF^V600E^ as a potential mechanistic target for their antiproliferative effects. Furthermore, the most potent EGFR inhibitors will be tested for their inhibitory effect against mutant-type EGFR (EGFR^T790M^), and the most potent anti-BRAF agents will be tested for their anticancer effect against the LOX-IMVI melanoma cell line, which has BRAF^V600E^ kinase overexpression. Finally, docking studies will be performed to investigate these compounds’ binding interactions within the active sites of target enzymes.

## 2. Results and Discussion

### 2.1. Chemistry

The synthesis of target compounds **3a–e**, **4a–c**, and **5a–c** is depicted in Figure 1. 5-chloro-3-formyl indole-2-carboxylate **1** [31] was reacted with amines **2a–e** [32] through reflux in ethanol followed by reduction of the intermediate imine with NaBH_4_ under reductive-amination conditions to yield secondary amines **3a–e** which subjected to saponification with LiOH to afford a carboxylic acids **4a–c**. The structures of compounds **3a–e** and **4a–c** were confirmed using ^1^H NMR, ^13^C NMR, and HRESI-MS spectroscopy (Varian Inova, University of Aberdeen, Meston Building, Aberdeen AB24 3UE, UK). ^1^H NMR spectrum of compound **3c** revealed the presence of a singlet signal δ 9.12 ppm of indole NH, the characteristic signals of ethyl group in the form of quartet at δ 4.33 ppm (2H) and triplet at δ 1.35 ppm (3H), a singlet signal at δ 4.18 ppm (2H) of CH_2_NH- group, and two triplets (each of 2H) at δ 2.88 and 2.74 ppm of NHCH_2_CH_2_ group. Moreover, the spectrum revealed the presence of the characteristic signals of both piperidine and aromatic protons. HRESI-MS *m/z* of **3c** calcd for [M + H]^+^ C_25_H_31_ClN_3_O_2_: 440.2099, found: 440.2100. The disappearance of the characteristic signals of the ethyl group in the form quartet at δ 4.33 ppm (2H) and triplet at δ 1.35 ppm (3H) and the appearance of a singlet signal at δ 3.43 ppm (2H) corresponding to COOH and NHCH_2_ characterize the ^1^H NMR spectrum of **4c**.

Intramolecular coupling of the carboxylic acids **4a–c** using BOP as the coupling reagent in the presence of DIPEA in DMF provided target compounds **5a–c**. ^1^H NMR spectrum of compound **5c** revealed the presence of a singlet signal δ 12.01 ppm of indole NH, a singlet siganl at δ 4.34 ppm (2H) of CH_2_N-group, and two triplets (each of 2H) at δ 3.66 and 2.79 ppm of NHCH_2_CH_2_ group. Furthermore, the disappearance of the singlet signal at 3.43 ppm (2H) corresponding to COOH and NHCH_2_ confirms cyclization.

### 2.2. Biology

#### 2.2.1. Cell Viability Assay

The viability test was performed on a normal human mammary gland epithelial cell line (MCF-10A). The MTT test was used to assess the viability of compounds **3a–e**, **4a–c**, and **5a–c** [33,34]. After four days of incubation with MCF-10A cells, the results showed that none of the substances tested were cytotoxic, and that the majority of those tested at 50 µM had cell viability levels greater than 87%.

#### 2.2.2. Antiproliferative Assay

Using the MTT assay [35,36] and erlotinib as the reference drug, compounds **3a–e**, **4a–c**, and **5a–c** were tested for antiproliferative efficacy against four human cancer cell lines: Panc-1 (pancreatic cancer cell line), MCF-7 (breast cancer cell line), HT-29 (colon cancer cell line), and A-549 (epithelial cancer cell line). The median inhibitory concentration (IC_50_) calculated (Graph Pad Software, San Diego, CA, USA) is shown in Table 1. For ease of manipulation, the average (GI_50_) versus the four cancer cell lines was used.

When compared to the reference drug erlotinib, which had a GI_50_ of 33 nM, compounds **3a–e**, **4a–c**, and **5a–c** all had substantial antiproliferative activity with GI_50_ values ranging from 29 nM to 78 nM. According to Table 1’s findings, **3a–e** were superior to **5a–c** and **4a–c** in their inhibitory actions against the tested cancer cell lines.

Compared to erlotinib (GI_50_ = 33 nM), the indole-2-carboxylate **3a–e** had the most antiproliferative effects, with GI_50_ values between 29 nM and 42 nM. Compound **3e** (R = *m*-piperidin-1-yl) was the most potent derivative, with a GI_50_ of 29 nM, outperforming the reference erlotinib, which had a GI_50_ of 33 nM. Compound **3e** was found to be more effective than erlotinib against Panc-1 (pancreatic cancer cell line), MCF-7 (breast cancer cell line), and A-549 (epithelial cancer cell line), Table 1. The substitution of the *m*-piperidine moiety in compound **3e** with the *p*-piperidine moiety in compound **3c** (R = *p*-piperidin-1-yl) resulted in a significant decrease in the antiproliferative activity of compound **3c**, which has a GI_50_ of 42 nM and is 1.5-fold less potent than **3e**, indicating the importance of the substitution position on the antiproliferative activity, where the *meta* position is better tolerated than the *para* one. Compound **3b** (R = *p*-pyrrolidin-1-yl) is the second most active antiproliferative compound, with a GI_50_ value of 31 nM, which is also higher than the reference compound erlotinib (GI_50_ = 33). Compound **3b** is more effective than erlotinib against the MCF-7 cancer cell line, with an IC_50_ value of 32 nM compared to 40 nM for erlotinib. With a mean GI_50_ value of 35 nM, the unsubstituted derivative **3a** (R = H) ranks third in activity against the four cancer cell lines and is even more potent than erlotinib against the MCF-7 cancer cell line, Table 1. The antiproliferative activity of the 2-methylpyrrolidine derivative **3d** (R = *p*-2-methylpyrrolidin-1-yl) was promising, with a GI_50_ of 38 nM, which is 1.3-fold less potent than **3e**. These findings demonstrated the importance of the substitution pattern on the phenyl group of the phenethyl moiety, with activity increasing in the order *m*-piperidine > *p*-pyrrolidine > H > *p*-2-methylpyrrolidine > *p*-piperidine.

Compounds **5a–c** had lower antiproliferative activity than compounds **3a–e**, with GI_50_ values of 48 nM, 62 nM, and 54 nM, respectively, compared to their congeners **3a–c**, which had GI_50_ values of 35 nM, 31 nM, and 42 nM, indicating that cyclization has a significant decrease in antiproliferative action.

Finally, the carboxylic acid derivatives **4a** (R = H), **4b** (R = *p*-pyrrolidin-1-yl), and **4c** (R = *p*-piperidin-1-yl) were the least potent against the four cancer cell lines, with GI_50_ values of 78 nM, 68 nM, and 72 nM, respectively, indicating the importance of the ethyl group at position two of indole nucleus for the antiproliferative action.

Panc-1 (pancreatic cancer cell line), MCF-7 (breast cancer cell line), HT-29 (colon cancer cell line), and A-549 (epithelial cancer cell line)

#### 2.2.3. EGFR Inhibitory Assay

The most effective antiproliferative derivatives **3a–e** were evaluated for their ability to inhibit EGFR using the EGFR-TK assay [37], and the findings are displayed in Table 2. The IC_50_ range for compounds **3a–e** inhibitions of EGFR were 68 to 89 nM. The *m*-piperidinyl derivative **3e** (R = *m*-piperidin-1-yl) was found to be the most potent of all synthesized derivatives, with an IC_50_ value of 68 nM, which was 1.2-fold more potent than erlotinib (IC_50_ = 80 nM). Compound **3b** (R = *p*-pyrrolidin-1-yl) is the second most active compound, with an IC_50_ value of 74 nM, and it is also more potent than erlotinib. Compounds **3a**, **3c**, and **3d** showed comparable inhibitory activity against EGFR to erlotinib, with IC_50_ values of 85, 89, and 82 nM, respectively. These results are consistent with the antiproliferative assay results and show that EGFR-TK is a possible target for the antiproliferative effects of compounds **3a–e**, and that compounds **3b** and **3e** were more potent against EGFR-TK than the reference erlotinib.

#### 2.2.4. BRAF^V600E^ Inhibitory Assay

The in vitro anti-BRAF^V600E^ activity of compounds **3a–e** was further investigated [38] using erlotinib and vemurafenib as reference compounds and results are shown in Table 2. The enzyme assay revealed that the five compounds tested significantly inhibited BRAF^V600E^, with IC_50_ ranges from 35 to 67 nM, Table 2. Interestingly, all the tested compounds **3a–e** had higher anti-BRAF^V600E^ activity than the reference erlotinib (IC_50_ = 60 nM) but were less potent than vemurafenib (IC_50_ = 30 nM). Once again, compounds **3b** (R = *p*-pyrrolidin-1-yl) and **3e** (R = *m*-piperidin-1-yl) had nearly the same inhibitory efficacy as BRAF^V600E^, with IC_50_ values of 39 and 35 nM, respectively, and were shown to be effective inhibitors of cancer cell proliferation (GI_50_ = 31 and 29 nM) as well as potent inhibitors of EGFR (IC_50_ = 74 and 68 nM). The unsubstituted derivative **3a** (R = H) demonstrated promising BRAF^V600E^ inhibitory activity, with an IC_50_ value of 43 nM, which was more potent than erlotinib but 1.4-fold less potent than vemurafenib. The findings of the study show that the tested compounds have potent antiproliferative activity and are effective at inhibiting both EGFR and BRAF^V600E^.

#### 2.2.5. EGFR^T790M^ Inhibitory Assay

The HTRF KinEASE-TK assay [39] was used to assess the inhibitory activity of the most potent compounds, **3b** and **3e**, against mutant-type EGFR (EGFR^T790M^). Osimertinib served as the positive control. As shown in Table 2, compounds **3b** and **3e** had excellent inhibitory activity against EGFR^T790M^, with IC_50_ values of 8.6 ± 2 and 9.2 ± 2 nM, respectively, equivalent to the reference osimertinib (IC_50_ = 8 ± 2 nM), which may explain their potent antiproliferative activity. Compounds **3b** and **3e** demonstrated 8-fold selectivity index toward EGFR^T790M^ protein over wild-type EGFR. These findings suggested that the phenethyl moiety’s *m*-piperidine and *p*-pyrrolidine substitutions are required for the inhibitory effect on EGFR^T790M^.

#### 2.2.6. LOX-IMVI Melanoma Cell Line Cytotoxicity Assay

The MTT assay was used to assess the anticancer activity of **3b** and **3e**, the most potent BRAF^V600E^ inhibitors, against the LOX-IMVI melanoma cell line, which has BRAF^V600E^ kinase overexpression [40,41]. The IC_50_ values of the test compounds were determined at 5-dose concentrations, with staurosporine serving as a control. Table 3 shows that the indole-2-carboxylate derivatives **3b** and **3e** have a high ability to reduce the viability of the LOX-IMVI cell line. Compound **3e** showed potent antiproliferative activity against the LOX-IMVI melanoma cell line with an IC_50_ value of 0.96 µM, followed by compound **3b** with an IC_50_ value of 1.12 µM in comparison to staurosporine (IC_50_ = 7.10 µM). The results of this assay add to the evidence that these compounds have potent antiproliferative activity as BRAF^V600E^ inhibitors.

### 2.3. Molecular Modeling

The most active antiproliferative compounds (**3a–e**) were subjected to in silico docking study in order to investigate their binding modes within BRAF^V600E^ active site. Molecular Operating Environment (MOE) software [42] was used as well as the crystal structure of the BRAF^V600E^ in complex with vemurafenib (PDB: 3OG7) [43]. The accuracy of docking simulation within the protein binding site was validated via redocking the co-crystallized ligand showing *S score* of −11.78 kcal/mol with *RMSD* value of 0.96 Å, (*S1*), Table 4. Docking score analysis of the examined compounds revealed that compounds **3b** and **3e** showed the highest negative scores (−10.12 and −10.40 kcal/mol, respectively) which are compatible with their in vitro BRAF^V600E^ inhibitory effects. Investigation of the compounds’ binding mode revealed that merely compound **3e** with (R = *m*-piperidin-1-yl) moiety extended comfortably along the large active site (Figure 2C,D). The compound probes the space of the active site in a manner analogous to that of the co-crystalized ligand, vemurafenib. The ligand 5-chloro-indole moiety stacks between the amino acid residues Trp531 and Phe583 inside the hydrophobic pocket forming pi-H interaction with Val471 (4.09 Å) as well as hydrophobic interactions with Trp531, Phe583, Cys532, Ile463, Thr592, and Val471. In addition, the chlorine atom forms halogen bond interaction with the key amino acid residue Cys532 (3.27 Å) at the site gate. Moreover, the ligand indole-2-carboxylate moiety forms ionic as well as H-bond interactions (3.13 Å) with the key amino acid Lys483. Additionally, the ligand stabilizes its complex within the active site by means of donating two H bond interactions with Thr529 (3.49Å), and Gly596 (3.41Å). On the other hand, the *para*-amino substitution in compounds **3b–d** did not allow the ligand to bind deeply inside the pocket compared with the *m*-piperidine moiety in compound **3e**. The latter finding confirms that the active site tolerates the *meta* substitution rather than the *para* one. Compound **3b** with (R = *p*-pyrrolidin-1-yl) forms multiple interactions although exhibiting another bent conformation within the active site relative to compound **3e**. The ligand indole-2-carboxylate moiety accepts a H-bond interaction from Lys483 (3.03 Å) as well as forming ionic interaction with Lys483 (3.03 Å). Moreover, the compound aminoethyl linker donates H bond to Thr529 (3.07 Å) while the (*p*-pyrrolidin-1-yl) moiety donates H-bond to Cys532 (3.28) at the gate of binding site. Furthermore, the (*p*-pyrrolidin-1-yl) moiety forms additional pi-H interaction with Cys532 (3.75 Å). (Figure 2A,B). The binding modes of compound **3c** and **3d** with R = *p*-piperidin-1-yl and R = *p*-2-methylpyrrolidin-1-yl, respectively, resemble that of compound **3b**, however they are neither interacting with Cys532 at the gate of active site nor Lys483 at the pocket hinge. Furthermore, the unsubstituted derivative **3a** probes the space of active site in an analogous pattern to that of compound **3e** while missing interactions with the amino acid residues Cys532, Thr531, and Val471 at the binding site. Other ligands interactions within the active site include hydrophobic ones with Phe583, Cys532, Thr529, Val471, Lys483, and Leu514.

Moreover, the most potent compounds **3b** and **3e** were subjected to docking study within the active pocket of the EGFR mutant type T790M (PDB: 5J9Z) [44]. The docking protocol was validated by redocking the co-crystallized ligand that exhibiting *S score* of −10.42 kcal/mol with RMSD value of 0.88 Å, (*S2*), Table 5. Compounds **3b** and **3e** exhibited comparable binding modes within the protein binding site (Figure 3). The ligand indole-2-carboxylate moiety binds deeply inside the hydrophobic pocket forming multiple hydrogen bond interactions with Met790 and Lys745 as well as pi-H interactions with Val726. In addition, the compounds form ionic bond interactions with Lys745 and Asp855. In addition, the *p*-pyrrolidin-1-yl moiety of compound **3b** forms ionic bond (3.74 and 3.62 Å) as well as H-bond interactions with Asp800 (3.62 Å) at the gate of the binding site. Moreover, the phenyl moiety of compound **3b** forms additional pi-H with Arg841 (4.82 Å). (Figure 3A,B). The ligand protein complexes are stabilized via other hydrophobic interactions with Asp800, Phe723, Leu844, Cys797, Leu718, Val726, Met790, and Lys745. Results of the docking simulations attributed to explaining the biological effects of the compounds **3a–e** relative to their binding affinity within the active site of BRAF^V600E^ as well as EGFR mutant type T790M and confirmed the dual kinase targets for the anti-proliferative activity of compounds **3b** and **3e**.

### 2.4. In Silico ADME/Pharmacokinetics Studies

The most active antiproliferative compounds **3a–e** were subjected to in silico ADME studies using the web tools SwissADME [45] as well as ADMETlab [46] by entering a list of two compounds’ SMILES (Simplified Molecule Input Line Entry Specification) provided by ChemDraw software. The in silico pharmacokinetic data (Table 6) showed that all compounds are orally active as they obey Lipinski’s rules of five with zero violation. All compounds are more likely to be a P-gp non-substrate. They exhibit high intestinal absorbance in the range of 88.9–90.5 %. They are capable of crossing BBB with logBB ranging from 0.22 to 0.31. According to Lipinski’s rules, logP should be ≤5. Thus, all compounds exhibited good permeability as indicated by logP values in the range of 4.23– 4.82. Regarding CYP inhibition, all compounds are considered inhibitors with probability exceeding 0.5 as shown in Table 7. The results predict that compounds **3a–e** exhibit good pharmacokinetics and ADME properties (Table 6 and Table 7).

## 3. Experimental

### 3.1. Chemistry

**General Details:** refer to Appendix A.

Compounds **1** [31] and **2a–e** [32] were prepared according to previously reported procedures.

#### 3.1.1. General Method for the Synthesis of Compounds **3a–e**

A mixture of compound **1** (0.73 g, 2.90 mmol, 1 equiv) and **2a–e** (1.2 equiv) in absolute ethanol (35 mL) was refluxed overnight with stirring. After cooling, NaBH_4_ (0.13 g, 1.2 equiv) was added portion wise over a period of 20 min with stirring for further 30 min at rt. H_2_O (30 mL) was added and the reaction mixture was concentrated in vacuo to a minimum, extracted with EtOAc (80 mL), dried over MgSO_4_, and concentrated in vacuo to give oil which was re-dissolved in EtOAc (30 mL) and treated with 3 M HCl till formation of white precipitate. The precipitate formed was filtered, washed with EtOAc, and dried to give secondary amine as its hydrochloride salt. The hydrochloride salt was dissolved in water/methanol 1:1 (70 mL) and treated with saturated solution of 5% NaOH till alkaline to liberate free amine. The resulting mixture was concentrated in vacuo to a minimum and extracted twice with EtOAc. The organic layer was dried under MgSO_4_, and evaporated under reduced pressure to give **3a–e**.

##### Ethyl 5-chloro-3-((phenethylamino)methyl)-1H-indole-2-carboxylate (3a)

Yield % 85; mp 104–105 °C. ν _max_ (KBr disc)/cm^−1^ 3296 (NH), 3063, 2950, 2845, 1693 (C=O), 1538, 1451, 1320, 1208, 1087, 895, 799, 748, 699. ^1^H NMR (400 MHz, Chloroform-*d*): δ 9.23 (s, 1H, indole NH), 7.65 (s, 1H, Ar-H), 7.25–7.12 (m, 8H, Ar-H, NHCH_2_), 4.29 (q, *J* = 6.8 Hz, 2H, OCH_2_CH_3_), 4.16 (s, 2H, CH_2_NHCH_2_CH_2_), 2.92 (q, *J* = 6.8 Hz, 2H, NHCH_2_CH_2_), 2.81 (t, *J* = 6.8 Hz, 2H, NHCH_2_CH_2_), 1.31 (t, *J* = 7.2 Hz, 3H, OCH_2_CH_3_). ^13^C NMR (101 MHz, Chloroform-*d*): δ 161.71 (C=O), 139.78, 133.90, 128.83, 128.63, 128.40, 126.18, 126.14, 126.09, 125.19, 121.31, 120.06, 112.93, 61.22, 50.22, 42.82, 36.18, 14.26. HRESI-MS *m/z* calcd for [M + H]^+^ C_20_H_22_ClN_2_O_2_: 357.1364, found: 357.1362.

##### Ethyl 5-chloro-3-((4-(pyrrolidin-1-yl)phenethylamino)methyl)-1H-indole-2-carboxylate (3b)

Yield % 74; mp 173–175 °C. ^1^H NMR (400 MHz, Chloroform-*d*) δ 7.68 (d, *J* = 1.9 Hz, 1H, Ar-H), 7.30–7.18 (m, 2H, Ar-H), 7.01 (d, *J* = 8.5 Hz, 2H, Ar-H), 6.48 (d, *J* = 8.5 Hz, 2H, Ar-H), 4.33 (q, *J* = 7.2 Hz, 2H, OCH_2_CH_3_), 4.19 (s, 2H, CH_2_NHCH_2_CH_2_), 3.28–3.20 (m, 4H, pyrrolidin-H), 2.90 (t, *J* = 7.1 Hz, 2H, NHCH_2_CH_2_), 2.75 (t, *J* = 7.1 Hz, 2H, NHCH_2_CH_2_), 2.02–1.94 (m, 4H, pyrrolidin-H), 1.36 (t, *J* = 7.1 Hz, 3H, OCH_2_CH_3_). ^13^C NMR (101 MHz, Chloroform-*d*) δ 161.79, 146.58, 133.95, 129.30, 128.88, 126.16, 126.11, 126.05, 125.26, 121.31, 120.15, 112.96, 111.75, 61.24, 50.68, 47.68, 42.84, 35.13, 25.43, 14.32. HRESI-MS *m/z* calcd for [M + H]^+^ C_24_H_29_ClN_3_O_2_: 426.1943, found: 426.1944.

##### Ethyl 5-chloro-3-((4-(piperidin-1-yl)phenethylamino)methyl)-1H-indole-2-carboxylate (3c)

Yield % 78; mp 75–78 °C. ν _max_ (KBr disc)/cm^−1^ 3312 (NH), 2932, 2850, 1705, 1612, 1540, 1514, 1451, 1380, 1236, 1131, 1024, 860, 800, 780. ^1^H NMR (400 MHz, Chloroform-*d*) δ 9.12 (s, 1H, indole NH), 7.70 (d, *J* = 1.8 Hz, 1H, Ar-H), 7.30–7.19 (m, 2H, Ar-H), 7.03 (d, *J* = 8.5 Hz, 2H, Ar-H), 6.84 (d, *J* = 8.6 Hz, 2H, Ar-H), 4.33 (q, *J* = 7.1 Hz, 2H, OCH_2_CH_3_), 4.18 (s, 2H, CH_2_NHCH_2_CH_2_), 3.13–3.05 (m, 4H, piperidin-H), 2.88 (t, *J* = 7.1 Hz, 2H, NHCH_2_CH_2_), 2.74 (t, *J* = 7.1 Hz, 2H, NHCH_2_CH_2_), 1.72–1.65 (m, 4H, piperidin-H), 1.58–1.52 (m, 2H, piperidin-H), 1.35 (t, *J* = 7.1 Hz, 3H, OCH_2_CH_3_). ^13^C NMR (101 MHz, Chloroform-*d*) δ 161.74, 150.73, 133.92, 130.52, 129.14, 128.96, 126.17, 126.08, 125.20, 121.74, 120.21, 116.77, 112.89, 61.20, 50.96, 50.50, 42.90, 35.37, 25.91, 24.28, 14.33. HRESI-MS *m/z* calcd for [M + H]^+^ C_25_H_31_ClN_3_O_2_: 440.2099, found: 440.2099.

##### Ethyl 5-chloro-3-((4-(2-methylpyrrolidin-1-yl)phenethylamino)methyl)-1H-indole-2-carboxylate (3d)

Yield % 75; mp 158–160 °C. ^1^H NMR (400 MHz, Chloroform-*d*) δ 9.25 (s, 1H, indole NH), 7.70 (d, *J* = 1.7 Hz, 1H, Ar-H), 7.24–2.23 (m, 2H, Ar-H), 7.01 (d, *J* = 8.5 Hz, 2H, Ar-H), 6.50 (d, *J* = 8.6 Hz, 2H, Ar-H), 4.33 (q, *J* = 7.1 Hz, 2H, OCH_2_CH_3_), 4.19 (s, 2H, CH_2_NHCH_2_CH_2_), 3.84–3.80, (m, 1H, pyrrolidin-H), 3.44–3.34 (m, 1H, pyrrolidin-H), 3.17–3.06 (m, 1H, pyrrolidin-H), 2.89 (t, *J* = 7.1 Hz, 2H, NHCH_2_CH_2_), 2.74 (t, *J* = 7.1 Hz, 2H, NHCH_2_CH_2_), 2.10–1.90 (m, 3H, pyrrolidin-H), 1.73–1.63 (m, 1H, pyrrolidin-H), 1.36 (t, *J* = 7.1 Hz, 3H, OCH_2_CH_3_), 1.15 (d, *J* = 6.2 Hz, 3H, CHCH_3_). ^13^C NMR (101 MHz, Chloroform-*d*) δ 161.83, 145.77, 133.97, 129.33, 128.96, 126.11, 126.09, 126.00, 125.24, 121.69, 120.21, 112.90, 111.81, 61.19, 53.66, 50.76, 48.32, 42.90, 35.24, 33.13, 23.32, 19.50, 14.31. HRESI-MS *m/z* calcd for [M + H]^+^ C_25_H_31_ClN_3_O_2_: 440.2099, found: 440.2092.

##### Ethyl 5-chloro-3-((3-(piperidin-1-yl)phenethylamino)methyl)-1H-indole-2-carboxylate (3e)

Yield % 73; oil. ^1^H NMR (400 MHz, Chloroform-*d*) δ 9.04 (s, 1H, indole NH), 7.71 (d, *J* = 1.8 Hz, 1H, Ar-H), 7.32–7.21 (m, 2H, Ar-H), 7.13 (t, *J* = 7.8 Hz, 1H, Ar-H), 6.80–6.69 (m, 2H, Ar-H), 6.62 (d, *J* = 7.5 Hz, 1H, Ar-H), 4.34 (q, *J* = 7.1 Hz, 2H, OCH_2_CH_3_), 4.19 (s, 2H, CH_2_NHCH_2_CH_2_), 3.13–3.05 (m, 4H, piperidin-H), 2.91 (t, *J* = 7.1 Hz, 2H, NHCH_2_CH_2_), 2.78 (t, *J* = 7.2 Hz, 2H, NHCH_2_CH_2_), 1.72–1.63 (m, 4H, piperidin-H), 1.59–1.52 (m, 2H, piperidin-H), 1.36 (t, *J* = 7.1 Hz, 3H, OCH_2_CH_3_). ^13^C NMR (101 MHz, Chloroform-*d*) δ 161.67, 152.40, 140.63, 133.89, 128.99, 128.97, 126.22, 126.13, 125.21, 121.73, 120.21, 119.51, 116.95, 114.28, 112.88, 61.22, 50.63, 50.34, 42.86, 36.68, 25.87, 24.31, 14.33. HRESI-MS *m/z* calcd for [M + H]^+^ C_25_H_31_ClN_3_O_2_: 440.2099, found: 440.2093.

#### 3.1.2. General Method for the Synthesis of Compounds **4a–c**

Compounds **3a–e** (0.68 mmol) in THF: H2O (5:1, 12 mL), LiOH (0.1 g, 4.09 mmol) was added. The reaction mixture was kept at 40 °C with stirring overnight. The residue after removal of solvent under reduced pressure was partitioned between Et_2_O/H_2_O (1:1) and the separated aqueous layer was acidified with 5% HCl. The formed precipitate was filtered and dried under vacuum to give **4a–c**.

##### 5-Chloro-3-((phenethylamino)methyl)-1H-indole-2-carboxylic acid (4a)

Yield % 87, mp 192–193 °C. ν _max_ (KBr disc)/cm^−1^ 3206 (br, OH and NH), 1693 (C=O), 1538, 1450, 1330, 1200, 1136, 802, 750, 699. ^1^H NMR (400 MHz, DMSO-*d*_6_) δ 11.92 (s, 1H, indole NH), 10.26 (s, 1H, COOH), 7.89 (d, *J* = 2.0 Hz, 1H, Ar-H), 7.44 (d, *J* = 8.7 Hz, 1H, Ar-H), 7.36–7.18 (m, 6H, Ar-H), 4.49 (s, 2H, CH_2_NHCH_2_CH_2_), 3.16 (t, *J* = 6.2 Hz, 2H, NHCH_2_CH_2_), 2.99 (t, *J* = 6.2 Hz, 2H, NHCH_2_CH_2_). ^13^C NMR (101 MHz, DMSO-*d*_6_) δ 164.11, 137.82, 133.76, 133.23, 129.06, 129.01, 128.49, 127.15, 124.84, 124.25, 119.43, 114.52, 108.74, 47.43, 41.07, 32.23. HRESI-MS *m/z* calcd for [M + H]^+^ C_18_H_18_ClN_2_O_2_: 329.1051, found: 329.1052.

##### 5-Chloro-3-((4-(pyrrolidin-1-yl)phenethylamino)methyl)-1H-indole-2-carboxylic acid (**4b**)

Yield % 85, mp 189–190 °C. ^1^H NMR (400 MHz, DMSO-*d*_6_) δ 11.04 (s, 1H, indole NH), 7.55 (s, 1H, Ar-H), 7.27 (d, *J* = 9.2 Hz, 1H, Ar-H),6.95 (m, 1H, Ar-H), 6.88 (d, *J* = 8.0 Hz, 2H, Ar-H), 6.34 (d, *J* = 8.4 Hz, 2H, Ar-H), 4.02 (s, 2H, CH_2_NHCH_2_CH_2_), 3.55 (t, *J* = 8.8 Hz, 2H, NHCH_2_CH_2_), 3.08 (t, *J* = 6.8 Hz, 4H, pyrrolidin-H), 2.45 (t, *J* = 2.0 Hz, 2H, NHCH_2_CH_2_), 1.87–1.82 (m, 4H, pyrrolidin-H). ^13^C NMR (101 MHz, DMSO-*d*_6_) δ 166.30, 146.76, 140.14, 136.77, 133.07, 130.16, 129.70, 127.25, 125.03, 123.34, 122.03, 113.90, 112.24, 47.99, 45.13, 40.33, 34.20, 25.54. HRESI-MS *m/z* calcd for [M + H]^+^ C_22_H_25_ClN_3_O_2_: 399.1635, found: 399.1638.

##### 5-Chloro-3-((4-(piperidin-1-yl)phenethylamino)methyl)-1H-indole-2-carboxylic acid (**4c**)

Yield % 90, mp 176–178 °C. ν _max_ (KBr disc)/cm^−1^ 3200 (br, OH and NH), 1690 (C=O), 1530, 1450, 1320, 1210, 1130, 802, 750, 699. ^1^H NMR (250 MHz, DMSO-*d*_6_): δ 11.32 (s, 1H, indole NH), 7.65 (s, 1H, Ar-H), 7.34 (d, *J* = 8.5 Hz, 1H, Ar-H), 7.15–6.90 (m, 3H, Ar-H), 6.79 (d, *J* = 3.5 Hz, 2H, Ar-H), 4.21 (s, 2H, CH_2_NHCH_2_CH_2_), 3.43 (s, 2H, OH, NHCH_2_), 3.02 (t, *J* = 4. 8 Hz, 4H, piperidin-H), 2.81 (t, *J* = 6. 8 Hz, 2H, NHCH_2_CH_2_), 2.68 (t, *J* = 6.5 Hz, 2H, NHCH_2_CH_2_), 1.65–1.39 (m, 6H, piperidin-H). ^13^C NMR (62.5 MHz, DMSO-*d*_6_): δ 165.3 (C=O), 150.2, 136.1, 132.6, 129.2, 129.0, 123.1, 122.0, 118.1, 116.1, 115.6, 113.6, 110.5, 49.9, 48.6, 40.9, 38.5, 33.5, 25.3. HRESI-MS *m/z* calcd for [M + H]^+^ C_23_H_27_ClN_3_O_2_: 412.1782, found: 412.1780.

#### 3.1.3. General Method for the Synthesis of Compounds **5a–c**

A mixture of **4a–c** (0.73 mmol, 1 equiv), BOP (0.45 g, 1.5 equiv), and DIPEA (0.24 mL, 2 equiv) in 20 mL DMF was stirred overnight at rt. The reaction mixture was diluted with EtOAc (50 mL) and successively washed with H_2_O, 5% HCl, saturated solution of NaHCO_3_, and finally with brine. The organic phase was dried over MgSO_4_ and concentrated under reduced pressure to yield a crude product which was purified by flash chromatography on silica gel using EtOAc/ hexane (2:1) as an eluent to give **5a–c**.

##### 7-Chloro-2-phenethyl-1,2-dihydropyrrolo [3,4-b]indol-3(4H)-one (**5a**)

Yield % 79, mp 245–247 °C. ν _max_ (KBr disc)/cm^−1^ 3148 (NH), 1659, 1451, 1320, 1250, 840, 808. ^1^H NMR (400 MHz, DMSO-*d*_6_) δ 12.04 (s, 1H, indole NH), 7.72 (d, *J* = 2.1 Hz, 1H, Ar-H), 7.45 (d, *J* = 8.8 Hz, 1H, Ar-H), 7.33–7.15 (m, 6H, Ar-H), 4.37 (s, 2H, CH_2_NCH_2_CH_2_), 3.75 (t, *J* = 7.3 Hz, 2H, NCH_2_CH_2_), 2.93 (t, *J* = 7.3 Hz, 2H, NCH_2_CH_2_). ^13^C NMR (101 MHz, DMSO-*d*_6_) δ 162.02, 139.96, 139.48, 136.57, 129.06, 128.83, 126.67, 124.85, 124.49, 124.08, 122.81, 119.65, 115.28, 46.45, 44.47, 34.80. HRESI-MS *m/z* calcd for [M + H]^+^ C_18_H_16_ClN_2_O: 311.0946, found: 311.0944.

##### 7-Chloro-2-(4-(pyrrolidin-1-yl)phenethyl)-1,2-dihydropyrrolo [3,4-b]indol-3(4H)-one (**5b**)

Yield % 75, mp 163–164 °C. ^1^H NMR (400 MHz, DMSO-*d*_6_) δ 11.99 (s, 1H, indole NH), 7.67 (d, *J* = 1.6 Hz, 1H, Ar-H), 7.39 (d, *J* = 8.8 Hz, 1H, Ar-H), 7.18 (dd, *J* = 8.8, 2.0 Hz, 1H, Ar-H), 6.98 (d, *J* = 8.4 Hz, 2H, Ar-H), 6.39 (d, *J* = 8.8 Hz, 2H, Ar-H), 4.29 (s, 2H, CH_2_NCH_2_CH_2_), 3.61 (t, *J* = 8.8 Hz, 2H, NCH_2_CH_2_), 3.10 (t, *J* = 6.4 Hz, 4H, pyrrolidin-H), 2.73 (t, *J* = 7.6 Hz, 2H, NCH_2_CH_2_), 1.88–1.84 (m, 4H, pyrrolidin-H). ^13^C NMR (101 MHz, DMSO-*d*_6_) δ 162.17, 146.94, 140.14, 136.91, 129.79, 125.75, 125.03, 124.63, 124.25, 123.03, 119.89, 115.49, 112.31, 47.93, 46.71, 45.13, 34.20, 25.56. HRESI-MS *m/z* calcd for [M + H]^+^ C_22_H_23_ClN_3_O: 398.1630, found: 398.1626.

##### 7-Chloro-2-(4-(piperidin-1-yl)phenethyl)-1,2-dihydropyrrolo [3,4-b]indol-3(4H)-one (**5c**)

Yield % 70, mp 207–209 °C. ν _max_ (KBr disc)/cm^−1^ 3150 (NH), 2936, 1655 (C=O), 1530, 1500, 1450, 1310, 1260, 1150, 842. ^1^H NMR (400 MHz, DMSO-*d*_6_) δ 12.01 (s, 1H, indole NH), 7.70 (d, *J* = 2.1 Hz, 1H, Ar-H), 7.43 (d, *J* = 8.8 Hz, 1H, Ar-H), 7.21 (dd, *J* = 8.8, 2.2 Hz, 1H, Ar-H), 7.05 (d, *J* = 8.5 Hz, 2H, Ar-H), 6.81 (d, *J* = 8.5 Hz, 2H, Ar-H), 4.34 (s, 2H, CH_2_NCH_2_CH_2_), 3.66 (t, *J* = 7.3 Hz, 2H, NCH_2_CH_2_), 3.07–2.99 (m, 4H, piperidin-H), 2.79 (t, *J* = 7.3 Hz, 2H, NCH_2_CH_2_), 1.62–1.42 (m, 6H, piperidin-H). ^13^C NMR (101 MHz, DMSO-*d*_6_) δ 161.99, 150.62, 139.95, 136.66, 129.49, 129.16, 124.83, 124.47, 124.05, 122.82, 119.65, 116.41, 115.28, 50.18, 46.46, 44.69, 33.92, 25.72, 24.32. HRESI-MS *m/z* calcd for [M + H]^+^ C_23_H_25_ClN_3_O: 394.1681, found: 394.1673.

### 3.2. Biology

#### 3.2.1. Cell Viability Assay and Evaluation of IC_50_

##### MTT Assay

The MTT assay was used to determine how the synthesized compounds affected the viability of mammary epithelial cells (MCF-10A) [33,34]. See Appendix A.

##### Antiproliferative Test

To investigate the antiproliferative potential of **3a–e**, **4a–c**, and **5a–c**, the MTT assay was carried out using various cell lines in accordance with previously reported procedures [35,36]. See Appendix A.

##### EGFR Inhibitory Assay

The EGFR-TK assay was used to evaluate the EGFR inhibitory effectiveness of **3a–e** [37]. See Appendix A.

##### BRAF Kinase Assay

The activity of **3a–e** against BRAF was investigated using a V^600E^ mutant BRAF kinase assay [38]. See Appendix A.

##### In Vitro Cytotoxicity of LOX-IMVI Melanoma Cell Line

The anticancer activity of the synthesized derivatives was determined using the MTT cytotoxicity assay on LOX-IMVI melanoma cell line [40,41]. See Appendix A.

## 4. Conclusions

A new series of 5-chloro-indole-2-carboxylate and pyrrolo [3,4-*b*]indol-3-one was synthesized and structurally characterized using various spectroscopic methods. The new compounds had no cytotoxic effects on human normal cell lines but demonstrated potent antiproliferative activities against four human cancer cell lines. Some of the compounds tested were found to be dual inhibitors of both wild type and mutant type EGFR and BRAF^V600E^. Molecular docking attempted to investigate the binding mode of the most active antiproliferative compounds **3a–e** within the binding site of BRAF^V600E^ in comparison with vemurafenib. Results proved that compound **3e**, with *m*-piperidinyl substitution at the phenethyl amine moiety, was found to fit more tightly within the active site than the other derivatives with *para*-amine substituents. Moreover, docking results of compounds **3b** and **3e** against EGFR^T790M^ concludes that the ligand indole-2-carboxylate scaffold binds intensely forming a combination of H-bond as well as hydrophobic interactions at the hydrophobic pocket of active site. In silico ADME and pharmacokinetic prediction revealed that compounds **3a–e** have good pharmacokinetic and ADME properties. Compounds **3b** and **3e** may act as anticancer agents targeting the EGFRT^790M^ and BRAF^V600E^ pathways after structural modifications, but more in vitro and in vivo testing is needed.

## Data Availability

The data will be provided upon request.

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
