# Peer review of "Design, Synthesis, and Antiproliferative Activity of New 5-Chloro-indole-2-carboxylate and Pyrrolo[3,4-*b*]indol-3-one Derivatives as Potent Inhibitors of EGFR^T790M^/BRAF^V600E^ Pathways"

_molecules, 2023, doi:10.3390/molecules28031269_

Round 1

Reviewer 1 Report

In this manuscript, the authors gave an attractive story about design, synthesis, and antiproliferative activity of new 5-chloro-indole-2-carboxylate and pyrrolo[3,4-b]indol-3-one derivatives as potent inhibitors of EGFRT790M/BRAFV600E pathways.  I would recommend this manuscript to be published. However, there are some issues that require improvement or clarification. 

1.       The SEM of the MTT results in table 1 was very high as compered to that in table 3 (MTT test also). Why there is such a discrepancy between the two results? Numbers after the dismal point should be added for table 1 and 2.

2.       Results in section 2.2.2. and table 1 were reported with IC50 values of nanomolar ranges. They must have been mistakenly reported in nanomolar ranges instead of micromolar ranges.

3.       Why the authors chose the four cell lines for their cytotoxicity evaluation? The correlation of these cells with protein kinases of this study should be demonstrated.

4.       The rational of the target drug design needs to be clearer.

5.       At end of the legend of table 1, add the names of the cell lines.

6.       The titles of all tests for section 4.2. should be added to the manuscript.

7.       In figure 1:

a.       Add the IC50 values for compounds I and II.

b.      Remove the IC50 vales of the antiproliferative assays of compounds V, VI and VII.

c.       For compounds 3a-e and 4a-c, there are two R substituents. One of them should be distinguished by a dash.

d.      For compounds 3a-e, 4a-c and 5a-c, remove 1a, 1b, 1c, 1d, 1e, 2a, 2b, 2c, 2d and 2e.

e.       In scheme one remove 1a, 1b, 1c, 1d and 1e since you do not have substituents on compound 1.

8.       In the subtitle 2.1. remove “is depicted in Scheme 1”

9.        Numbers following authors’ names and affiliations need to be corrected.

Author Response

Reviewer #1

Comments and Suggestions for Authors

In this manuscript, the authors gave an attractive story about design, synthesis, and antiproliferative activity of new 5-chloro-indole-2-carboxylate and pyrrolo[3,4-b]indol-3-one derivatives as potent inhibitors of EGFRT790M/BRAFV600E pathways.  I would recommend this manuscript to be published. However, there are some issues that require improvement or clarification. 

  1. The SEM of the MTT results in table 1 was very high as compared to that in table 3 (MTT test also). Why there is such a discrepancy between the two results? Numbers after the dismal point should be added for table 1 and 2.

Response

We appreciate the reviewer's input, but Tables 1 and 2 provide IC50 values in the nanomolar range, whereas Table 3 shows values in the micromolar range. Tables 1 and 2's SEM values are all within the allowed range.

  1. Results in section 2.2.2. and table 1 were reported with IC50 values of nanomolar ranges. They must have been mistakenly reported in nanomolar ranges instead of micromolar ranges.

Response

No, sir. Because most of the tested compounds are potent, the IC50 values (Tables 1 and 2) quoted are in the nanomolar range, not the micromolar range.

  1. Why the authors chose the four cell lines for their cytotoxicity evaluation? The correlation of these cells with protein kinases of this study should be demonstrated.

Response

We thank the reviewer for his comment. These four cancer cell lines demonstrated overexpression of EGFR. Please refer to our previously published manuscript (Chem Biol Drug Des. 90 (2107) 443-449, European Journal of Medicinal Chemistry 135 (2017) 34-48, Bioorganic chemistry 76 (2018) 314-325, European Journal of Medicinal Chemistry  146 (2018) 260-273, European Journal of Medicinal Chemistry 2018, 156: 774-789, European Journal of Medicinal Chemistry 2019; 176: 117-128, European Journal of Medicinal Chemistry 2019;  177: 1-11, Bioorganic Chemistry 2019; 89: 102997, Bioorganic Chemistry, 2020; 102, 104090, Bioorganic Chemistry 106 (2021) 104510, Bioorganic Chemistry 120 (2022) 105616). Moreover, when the compounds showed a potent BRAFV600E inhibitory action, we test our compounds against LOX-IMVI melanoma cell line, which has BRAFV600E kinase overexpression. Please refer to reference [40, 41] in this manuscript.

  1. The rational of the target drug design needs to be clearer.

Response

Simply put, we are interested in the development of small molecule inhibitors as multi-targeted cancer chemotherapy, so we have published numerous articles, some of which are referred to in the introduction. More information about the strategy can be found in these references.

  1. At end of the legend of table 1, add the names of the cell lines.

Response

Done as advised.

  1. The titles of all tests for section 4.2. should be added to the manuscript.

Response

Done as advised.

  1. In figure 1:
    a. Add the IC50 values for compounds I and II.

Response

Done as advised. For compound I not reported as the activity was assessed through Western Blot.

b. Remove the IC50 vales of the antiproliferative assays of compounds V, VI and VII.

Response

Done as advised.

c. For compounds 3a-e and 4a-c, there are two R substituents. One of them should be distinguished by a dash.

Response

Done as advised.

d. For compounds 3a-e, 4a-c and 5a-c, remove 1a, 1b, 1c, 1d, 1e, 2a, 2b, 2c, 2d and 2e.

Response

Done as advised.

e. In scheme one remove 1a, 1b, 1c, 1d and 1e since you do not have substituents on compound 1.

Response

Done as advised.

8. In the subtitle 2.1. remove “is depicted in Scheme 1”

Response

Done as advised. It has been modified.

  1. Numbers following authors’ names and affiliations need to be corrected.

Response

Done as advised.

Reviewer 2 Report

Manuscript Title: Design, synthesis, and anti-proliferative activity of new 5-chloro-indole-2-carboxylate and pyrrolo [3,4-b]indol-3-one derivatives as potent inhibitors of EGFRT790M/BRAFV600E pathways

Manuscript Number: molecules-2180230

Article Type: Full Paper                                                                                                            

Comments:

Overall, the manuscript “Design, synthesis, and antiproliferative activity of new 5-chloro-indole-2-carboxylate and pyrrolo[3,4-b]indol-3-one derivatives as potent inhibitors of EGFRT790M/BRAFV600E pathways” by Bahaa G. M. Youssif and co-workers nicely executed the work of 5-chloro-indole-2-carboxylate and pyrrolo[3,4-b]indol-3-one derivatives as potent inhibitors of mutant EGFR/BRAF pathways are thought to be crucial targets for the development of anticancer drugs with enhanced biological data. The authors developed of a novel series of 5-chloro-indole-2-carboxylate 3a-e, 4a-c, and pyrrolo[3,4-b]indol-3-ones 5a-c derivatives as potent inhibitors and showing antiproliferative activity. The cell viability assay data of most of the compounds revealed that none of the compounds were cytotoxic and that the majority of those tested at 50 µM had cell viability levels greater than 87%. The panel of compounds 3a-e tested for EGFR inhibition, with IC50 values ranging from 68 nM to 89 nM. The most potent derivative was found to be the m-piperidinyl derivative 3e, with an IC50 value of 68 nM, which was 1.2-fold more potent than erlotinib (IC50 = 80 nM).  Additionally, molecular docking of 3a and 3b against BRAFV600E and EGFRT790M enzymes revealed high binding affinity and active site interactions compared to the co-crystalized ligands. Further, the pharmacokinetics properties (ADME) of 3a-e revealed safety and good pharmacokinetic profile

I thoroughly enjoyed reading the whole manuscript. This work was well performed with good supportive experimental data and nicely drafted this manuscript without grammatical errors. I am strongly recommending this manuscript without any revision before going for publication. For improving the quality of work, here I am recommending some suggestions and comments for future studies.

Modest items that need to be addressed include:

1.       It would be interesting if you do the broader SAR on the replacement of chlorine substitution and substitutions on indole ‘-NH’

2.       The IC50 of compounds 3a-e, 4a-c, and 5a-c are most consistent with the antiproliferative activity (range 30-75 nM). Can you comment on this?

3.       I recommend you go for in vivo studies for some of the lead compounds.  

Author Response

Reviewer #2

Comments and Suggestions for Authors

Manuscript Title: Design, synthesis, and anti-proliferative activity of new 5-chloro-indole-2-carboxylate and pyrrolo [3,4-b]indol-3-one derivatives as potent inhibitors of EGFRT790M/BRAFV600E pathways

Manuscript Number: molecules-2180230

Article Type: Full Paper                                                                                                            

Comments:

Overall, the manuscript “Design, synthesis, and antiproliferative activity of new 5-chloro-indole-2-carboxylate and pyrrolo[3,4-b]indol-3-one derivatives as potent inhibitors of EGFRT790M/BRAFV600E pathways” by Bahaa G. M. Youssif and co-workers nicely executed the work of 5-chloro-indole-2-carboxylate and pyrrolo[3,4-b]indol-3-one derivatives as potent inhibitors of mutant EGFR/BRAF pathways are thought to be crucial targets for the development of anticancer drugs with enhanced biological data. The authors developed of a novel series of 5-chloro-indole-2-carboxylate 3a-e, 4a-c, and pyrrolo[3,4-b]indol-3-ones 5a-c derivatives as potent inhibitors and showing antiproliferative activity. The cell viability assay data of most of the compounds revealed that none of the compounds were cytotoxic and that the majority of those tested at 50 µM had cell viability levels greater than 87%. The panel of compounds 3a-e tested for EGFR inhibition, with IC50 values ranging from 68 nM to 89 nM. The most potent derivative was found to be the m-piperidinyl derivative 3e, with an IC50 value of 68 nM, which was 1.2-fold more potent than erlotinib (IC50 = 80 nM).  Additionally, molecular docking of 3a and 3b against BRAFV600E and EGFRT790M enzymes revealed high binding affinity and active site interactions compared to the co-crystalized ligands. Further, the pharmacokinetics properties (ADME) of 3a-e revealed safety and good pharmacokinetic profile.

I thoroughly enjoyed reading the whole manuscript. This work was well performed with good supportive experimental data and nicely drafted this manuscript without grammatical errors. I am strongly recommending this manuscript without any revision before going for publication. For improving the quality of work, here I am recommending some suggestions and comments for future studies.

First of all, we would like to thank the reviewer for accepting our work for publication.

Modest items that need to be addressed include:

  1. It would be interesting if you do the broader SAR on the replacement of chlorine substitution and substitutions on indole ‘-NH’

Response

We completely agree with the comment of the respected reviewer. We previously investigated the effect of 4,5, and 6-position substitutions of the indole ring with various electron withdrawing groups such as halogens and trifluoromethyl groups, as well as electron donating groups such as methyl, and discovered that 5-chloro is the most tolerated derivative, which we use in the current manuscript, please refer to reference (25-27). We are currently preparing some N-alkyl indole derivatives for future publication.

  1. The IC50 of compounds 3a-e, 4a-c, and 5a-c are most consistent with the antiproliferative activity (range 30-75 nM). Can you comment on this?

Response

Sorry sir, but did you mean the following paragraph? (Compounds 3a, 3c, and 3d showed comparable inhibitory activity against EGFR to erlotinib, with IC50 values of 85, 89, and 82 nM, respectively. These results are consistent with the antiproliferative assay results and show that EGFR-TK is a possible target for the antiproliferative effects of compounds 3a–e, and that compounds 3b and 3e were more potent against EGFR-TK than the reference erlotinib). If so, the results of the EGFR inhibitory assay matched those of the antiproliferative assay, with the most potent derivatives as EGFR inhibitors also being the most potent derivatives as antiproliferative agents.

  1. I recommend you go for in vivostudies for some of the lead compounds.  

Response

Yes, sir, as stated in the conclusion, we have already submitted two compounds for in vivo testing.

Reviewer 3 Report

The manuscript by Lamya and co-autors submitted for review is interesting, but the authors should make some improvements.

The authors should remove compounds 1 and 2 from the description of compounds 3 and 4 in Figure 1.

I think that giving an average IC50 does not make sense. Authors should remove this column from Table 1.

Why did the authors use Staurosporine as a reference in studies on LOXIMVI cancer cells?

If the authors consider it right to present the results of in silico research, they should do it better, in a wider scope and with the use of software that gives specific numerical values - creating a yes/no table (Table 7) does not make sense. In the case of CYP inhibition, this can be done using the Percepta program. If there is no such possibility, it is enough to describe it in one sentence and put it in the table.

The authors should add HR MS spectra to the supplement as well as 1H NMR spectra to present in the entire range, in some spectra there is no aliphatic range.

Editing also needs improvement (justification, subscripts, DMSO-not dmso).

Author Response

Reviewer #3

Comments and Suggestions for Authors

The manuscript by Lamya and co-authors submitted for review is interesting, but the authors should make some improvements.

The authors should remove compounds 1 and 2 from the description of compounds 3 and 4 in Figure 1.

Response

Done as advised.

I think that giving an average IC50 does not make sense. Authors should remove this column from Table 1.

Response

To facilitate data manipulation, we use mean IC50 (GI50) for each compound against the four cancer cell lines. We are unable to remove the GI50 because it was used in the discussion section; please accept our apologies.

Why did the authors use Staurosporine as a reference in studies on LOXIMVI cancer cells?

Response

In each test, we try to use the best available reference compound. For LOX-IMVI melanoma cell line cytotoxicity assay the reported reference is Staurosporine not Erlotinib or Vemurafenib, so we follow the reported procedure ( Bioorganic Chemistry 120 , 2022,105616).

If the authors consider it right to present the results of in silico research, they should do it better, in a wider scope and with the use of software that gives specific numerical values - creating a yes/no table (Table 7) does not make sense. In the case of CYP inhibition, this can be done using the Percepta program. If there is no such possibility, it is enough to describe it in one sentence and put it in the table.

Response

Done as advised. The ADME section has been updated.

The authors should add HR MS spectra to the supplement as well as 1H NMR spectra to present in the entire range, in some spectra there is no aliphatic range.

Response

Done as directed. HRMS spectra have been added to the supplementary file. Moreover, we may provide more than one 1H NMR spectrum for the same compound in order to focus on a specific area, but for all compounds, we provide the detailed 1H NMR spectrum in the experimental section..

Editing also needs improvement (justification, subscripts, DMSO-not dmso).

Response:

The whole manuscript was revised, and all necessary corrections have been done.

Round 2

Reviewer 3 Report

I accept the Authors' answers. Please make sure to post the final version of the supplement as there are no HR MS spectra in this one.